# Strength and Durability Characteristics of Cement Composites with Recycled Water and Blast Furnace Slag Aggregate

**DOI:** 10.3390/ma14092156

**Published:** 2021-04-23

**Authors:** Se-Jin Choi, Sung-Ho Bae, Jae-In Lee, Ji-Hwan Kim

**Affiliations:** Department of Architectural Engineering, Wonkwang University, 460 Iksan-daero, Iksan 54538, Korea; caos1344@naver.com (S.-H.B.); wodls103@naver.com (J.-I.L.); 3869kjh@naver.com (J.-H.K.)

**Keywords:** recycled water, blast furnace slag aggregate, strength, cement composite, drying shrinkage, carbonation

## Abstract

Recently, interest in sustainable development has been increased. In this regard, efforts have been made to prevent environmental pollution, and research on the recycling of construction industry byproducts has been actively conducted in the construction industry. In South Korea, about 20 million tons of waste wash water from the ready-mixed concrete production process are generated, and some of them are recycled using recycling facilities in a ready-mixed concrete plant, but a significant portion of them is discharged or landfilled without permission, causing environmental problems. To increase the recycling rate of steel slag and reduce environmental pollution in the construction industry, we simultaneously applied blast furnace slag fine aggregate (BSFA) and recycled water (RW) to cement mortar. In this study, to examine the feasibility of RW and BSFA, we evaluated the fluidity, compressive strength, tensile strength, drying shrinkage, carbonation depth, and chloride penetration resistance of cement mortar using RW and BSFA. From the test results, the 28-day compressive strengths of all samples using RW and BSFA were higher than that of the control sample. In the case of samples using RW, as the BSFA replacement ratio was increased, the carbonation depth of the samples decreased. Therefore, when RW and BSFA are used properly, the mechanical properties of cement mortar, carbonation resistance, and chloride ion penetration resistance are expected to be effectively improved.

## 1. Introduction

Recently, interest in sustainable development has increased. In addition, owing to the immense growth of the global concrete industry, the shortage of natural aggregates has emerged as a serious problem.

In this regard, efforts have been made to prevent environmental pollution, and research on the recycling of construction industry byproducts has been actively conducted in the construction industry [1,2,3,4]. Among the byproducts of the construction industry, about 20 million tons of waste wash water from the ready-mixed concrete production process are generated in Korea, and some of them are recycled using recycling facilities in a ready-mixed concrete plant, but a significant portion of them is discharged or landfilled without permission, causing environmental problems. Waste wash water is divided into recycled water (RW), sludge water, supernatant water, and sludge contents [5], and various studies have reported using recycled wash water.

Franco et al. [6] investigated the recycling of waste wash water generated in a ready-mixed concrete plant, and as a result, using RW in concrete was found to reduce concrete capillary water absorption and improve durability.

Yang et al. [7] reviewed the durability and mechanical properties of concrete using RW and reported that mixing up to 8% of sludge contents did not affect freezing and thawing resistance.

Ekolu et al. [8] investigated the feasibility of RW that satisfies the ASTM C 94 standards and reported that the compressive strength of mortar increased with the mixing ratio of RW.

Oh et al. [9] evaluated the characteristics of lightweight aggregate mortar using RW as prewetting water for artificial lightweight aggregates. Based on the result of the experiment, it was reported that the artificial lightweight aggregate mortar using RW with a sludge content of 5% showed higher compressive strength and carbonation resistance than the control sample.

Meanwhile, studies on alternative aggregate resources were conducted owing to the recent shortage of natural aggregates for concrete [10,11], and some studies on using blast furnace slag, a byproduct of the steel industry, as aggregates for concrete were reported [12,13,14,15].

Park et al. [11] reviewed the utility of coal gasification slag (CGS) as a fine aggregate for concrete and reported that, based on its chemical composition, CGS generally satisfies the criteria posed by the standards concerning the number of harmful substances, except for SO_3_.

Valcuende et al. [13] evaluated the shrinkage characteristics of self-compacting concrete using blast furnace slag aggregate and reported that autogenous and drying shrinkage increased with the amount of slag.

However, studies on mortar or concrete using steel slag aggregate and RW were not reported. To increase the recycling rate of steel slag and reduce environmental pollution in the construction industry, we intend to apply blast furnace slag fine aggregate (BSFA) and RW to cement mortar simultaneously. In the literature, it is reported that the high alkalinity of RW can improve the reactivity of blast furnace slag powder [16]. However, if blast furnace slag is used for cement composites in the aggregate form rather than in the powder form, the characteristics of cement mortar are expected to be different.

This study examines the feasibility of RW and BSFA in cement mortar, the fluidity, compressive strength, tensile strength, drying shrinkage, carbonation depth, and chloride penetration resistance of cement mortar using RW and BSFA were evaluated. This research, including RW and BSFA, brings novelty to the eco-friendly construction materials research field.

## 2. Materials and Methods

### 2.1. Materials

The cement used in this study was ordinary Portland cement (OPC), manufactured by Asia Co. (Seoul, Korea). As fine aggregates, natural sand with a specific gravity of 2.6 and a fineness modulus of 2.45 and a BSFA with a specific gravity of 2.8 and a fineness modulus of 2.3 generated in the form of fine aggregates from P Company (Seoul, Korea) was used. In the case of RW, by referring to the previous literature [9], sludge with a ratio of cement and sand fines (lower than 0.15 mm) of 4:1 was prepared (Table 1), and the sludge content was fixed at 5% in RW. Thereafter, RW with a sludge content of 5% was used as the mixing water. Table 2 and Table 3 show the chemical composition and physical properties of the cement and fine aggregates used in this study. Figure 1 shows the shape and SEM image of BSFA. Figure 2 shows the particle size distribution of each aggregate.

### 2.2. Mixing Proportions and Specimen Preparation

Table 4 shows the mixture proportion of cement mortar. The water-cement ratio was fixed at 50%, and the sludge content of RW was 5%, which showed good characteristics in a previous study [17]. RW was used as the mixing water. A mixture of cement mortar without RW and BSFA was used as the control mixture. In the case of cement mortar samples using RW, 0, 10, 20, 30, and 40% of BSFA were replaced by the volume of the fine aggregate. In this study, we did not use any chemical admixture. Further, 50 mm cube specimens were prepared for a compressive strength test, 50 × 100 mm^2^ cylindrical specimens were prepared for a split-tensile strength test, 40 × 40 × 160 mm^3^ specimens were prepared for drying shrinkage and carbonation tests, and 100 × 50 mm^2^ specimens were prepared for a chloride-ion penetration test. Subsequently, we demolded the specimens after 24 h and cured them in a water tank at 20 °C until the required age.

Mortar flow and compressive strength were measured according to KS L 5105 [18], and tensile strength was measured according to KS F 2423 [19]. The presented strength test values are the average values of three samples.

Drying shrinkage was measured using a contact gauge, as shown in Figure 3, according to KS F 2424 [20]. For the carbonation test, the carbonation depth was measured using a phenolphthalein solution after the carbonation process in an accelerated carbonation chamber according to KS F 2584 [21]. In addition, the chloride-ion penetration test was performed according to ASTM C 1202 [22].

## 3. Results and Discussion

### 3.1. Mortar Flow

Figure 4 shows the change in the mortar flow concerning the replacement ratio of BSFA. As shown from the figure, the flow values of the control mixture and RW5BS0 mixture without BSFA were both equal to approximately 184 mm. The flow of the RW5BS40 sample using 40% of BSFA was approximately 203 mm, which was approximately 12% higher than that of the control sample. In this study, using RW did not significantly affect the mortar flow, and the mortar flow increased with the replacement ratio of BSFA, which seems to be due to the vitreous properties of blast furnace slag aggregates.

### 3.2. Compressive Strength

Figure 5 shows the compressive strength of the mortar samples with RW concerning the replacement ratio of BSFA. As shown in the figure, the 7-day compressive strength of the control sample was approximately 38.5 MPa, and the compressive strengths of the RW5BS0 and RW5BS10 samples using RW and 0 and 10% of BSFA were approximately 4–7% lower than that of the control sample. In contrast, the 7-day compressive strength of the sample using RW and 20% or more of BSFA was approximately from 38.3 to 39.5 MPa, respectively, similar to or somewhat higher than that of the control sample. In the case of an age of 28 days, the compressive strengths of the control sample and the RW5BS0 sample were similar, and the compressive strengths of all the samples using RW and BSFA were higher than that of the control sample.

In particular, the compressive strength of the RW5BS40 sample using RW and 40% of BSFA was approximately 52.7 MPa, which was approximately 18% higher than that of the RW5BS0 sample using RW and not using BSFA. This tendency was similar at the age of 56 days; that is, the compressive strength of the control sample was the lowest at 60.5 MPa, and the compressive strength of the RW5BS0 sample using RW was approximately 4% higher than that of the control sample. In addition, as the replacement ratio of BSFA was increased, the compressive strength of the samples was higher.

The 56-day compressive strength of the sample using 10–40% BSFA and RW were approximate 3–8% higher than that of the RW5BS0 sample.

Therefore, when RW and BSFA were used simultaneously, the compressive strength of the cement mortar improved, which is likely because the high alkalinity of the RW improved the reactivity of the BSFA [23,24,25].

### 3.3. Split-Tensile Strength

Figure 6 shows the 28-day split-tensile strength of the mortar sample using RW concerning the BSFA replacement ratio. As shown in the figure, the tensile strength of the control sample was the lowest at approximately 3.73 MPa. In the case of the split-tensile strength, similar to the tendency of the compressive strength, the split-tensile strength of the samples increased with the BSFA replacement ratio. The split-tensile strengths of the RW5BS30 and RW5BS40 samples using more than 30% of BSFA were 3.93 MPa and 4.24 MPa, respectively, which were about 5–14% higher than that of the control sample.

In general, the tensile strength of the samples using BSFA was relatively higher than that of the control sample, but the ratio of tensile strength to compressive strength was approximately 8.0–8.4% regardless of the BSFA use.

### 3.4. Drying Shrinkage

Figure 7 shows the change in the drying shrinkage of the mortar samples with RW and BSFA.

After an age of 56 days, the drying shrinkage of the control sample was approximately 0.142%, and the drying shrinkage of the RW5BS0 sample using RW without BSFA was 0.133%, which was lower than that of the control sample. In the case of the mixtures using RW and BSFA simultaneously, the drying shrinkages of the RW5BS10 and RW5BS20 samples using 10 and 20% of BSFA were approximately 0.128–0.130%, which was relatively lower than that of the control sample. The drying shrinkages of the RW5BS30 and RW5BS40 samples using 30 and 40% of BSFA were approximately 0.142–0.144%, similar to the control sample. Therefore, in this study, it was found that when the replacement ratio of BSFA was lower than 20%, the drying shrinkage of cement mortar using RW was effectively reduced.

### 3.5. Accelerated Carbonation Depth

Figure 8 shows the carbonation depth of the samples using RW concerning the BSFA replacement ratio after 28 days of the accelerated carbonation test. As shown in the figure, the carbonation depth of the control sample was approximately 0.97 mm, which was the largest, and the carbonation depth of the RW5BS0 sample using RW without BSFA was approximately 0.86 mm, which was approximately 11% lower than that of the control sample. In the case of the samples using RW, as the BSFA replacement ratio was increased, the carbonation depth of the samples decreased, and the accelerated carbonation depth of the RW5BS40 sample using RW and 40% of BSFA was the lowest at approximately 0.56 mm. In general, as the compressive strength of the sample was increased, the carbonation depth tended to decrease.

### 3.6. Chloride-Ion Penetrability

Figure 9 shows the variation in the chloride-ion penetrability of the sample concerning the BSFA replacement ratio after an age of 28 days. The total charge passed through the sample was calculated according to ASTM C 1202. The charge passed through the control sample was approximately 9865 C, and the charge passed through the RW5BS0 sample using RW without BSFA was approximately 11,381 C, which was approximately 15% higher than that of the control sample. However, in the mixtures using RW and BSFA, the charge passed through the sample tended to decrease gradually as the BSFA replacement ratio was increased. The charge passed through the samples using more than 20% of BSFA was approximately from 6667 to 9731 C, which was lower than the charge passed through the control sample. In particular, in the RW5BS30 sample using 30% of BSFA and RW, the charge passed through the sample was approximately 6667 C, which was approximately 32% lower than that of the control sample. Therefore, when only RW is used in cement mortar, it may be detrimental to its chloride ion penetration resistance, but the simultaneous use of BSFA and RW in cement mortar is expected to increase the chloride ion penetration resistance of the sample.

## 4. Conclusions

The conclusions of this study can be summarized as follows:(1)In this study, using RW did not significantly affect the mortar flow, but the mortar flow increased with the replacement ratio of BSFA, which seems to be due to the vitreous properties of blast furnace slag aggregates;(2)The 28-day compressive strengths of the control sample and RW5BS0 sample were similar, and the compressive strengths of all the samples using RW and BSFA were higher than that of the control sample. Further, the split-tensile strengths of the samples increased with the BSFA replacement ratio, similar to the tendency of the compressive strength;(3)It was found that when the replacement ratio of BSFA was lower than 20%, the drying shrinkage of the cement mortar using RW was effectively reduced. In the case of the samples using RW, as the BSFA replacement ratio was increased, the carbonation depth of the samples decreased, and the accelerated carbonation depth of the RW5BS40 sample using RW and 40% of BSFA was the lowest at approximately 0.56 mm;(4)In the case of the mixtures using RW and BSFA, the charge passed through the sample tended to decrease gradually as the BSFA replacement ratio was increased;(5)Therefore, when RW and BSFA are used properly, it is expected that the mechanical properties of cement mortar, carbonation resistance, and chloride ion penetration resistance can be enhanced.

However, further studies are required to establish the strength development mechanism and the relationship between chloride penetration resistance, durability and chemical composition of materials, water-binder ratio, and alkali content, among other factors.

## Figures and Tables

**Figure 1 materials-14-02156-f001:**
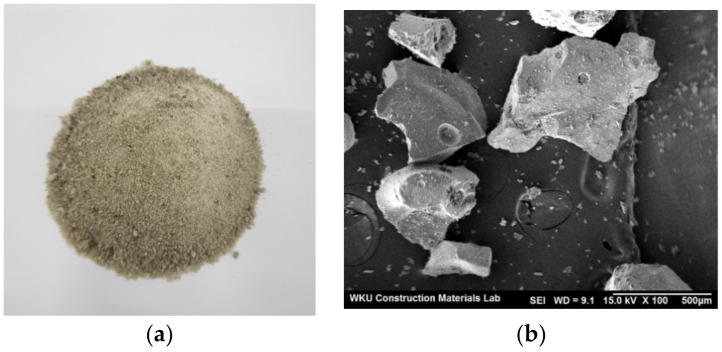
Blast furnace slag fine aggregate: (**a**) shape, (**b**) SEM image

**Figure 2 materials-14-02156-f002:**
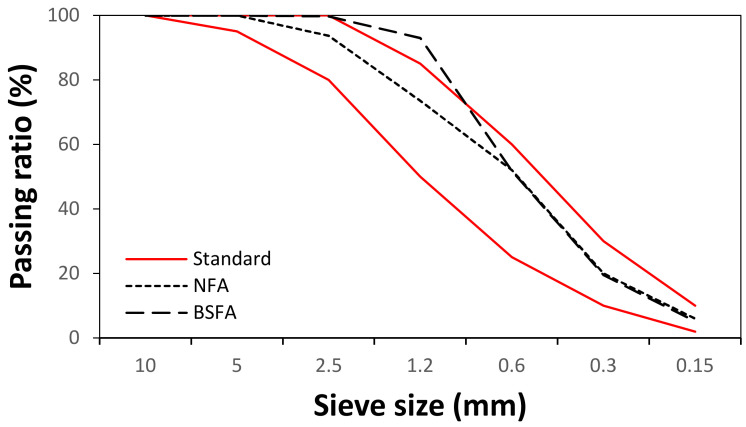
Particle size distribution of fine aggregate.

**Figure 3 materials-14-02156-f003:**
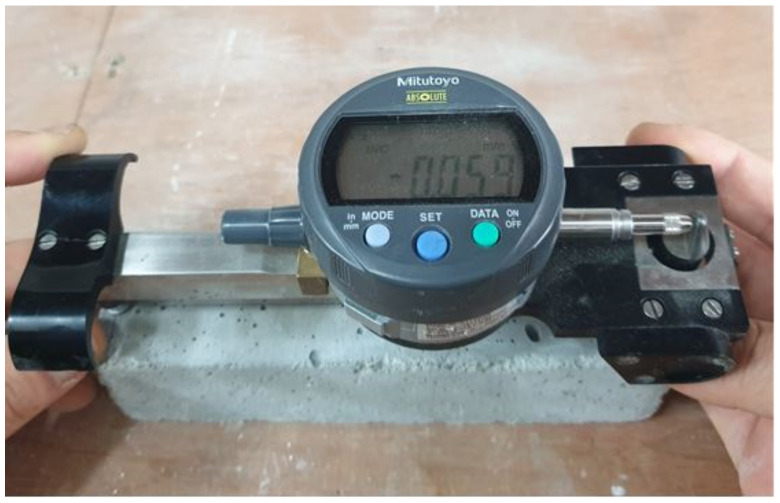
Measurement of drying shrinkage.

**Figure 4 materials-14-02156-f004:**
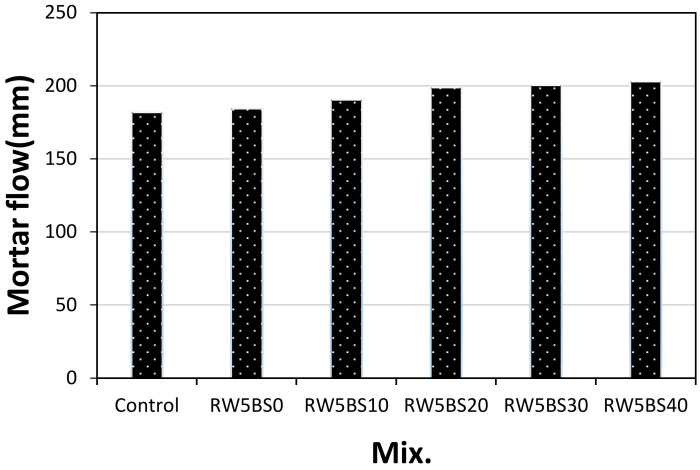
Mortar flow.

**Figure 5 materials-14-02156-f005:**
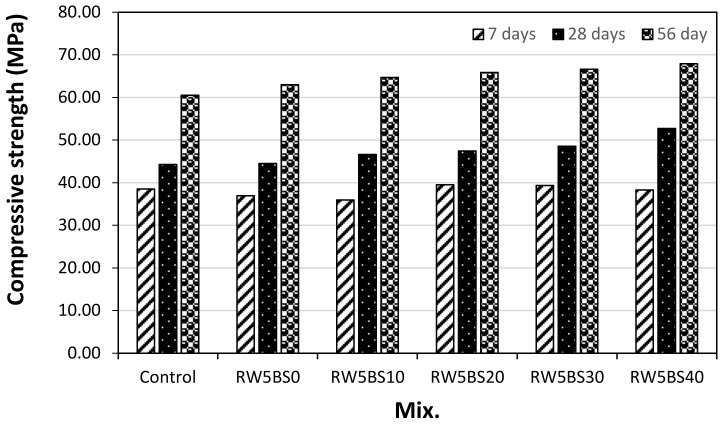
Compressive strength.

**Figure 6 materials-14-02156-f006:**
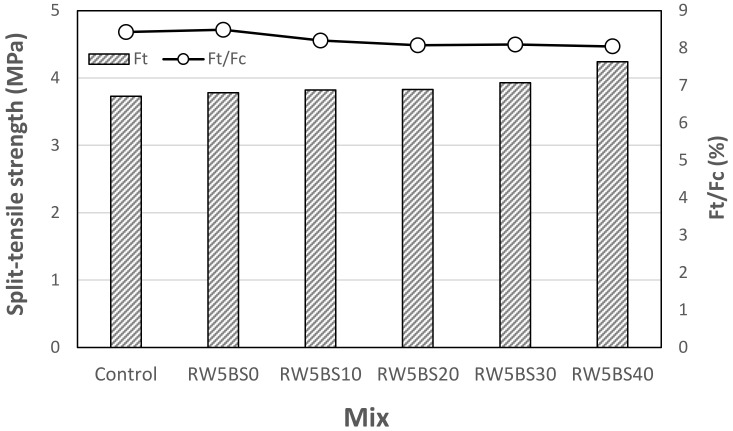
Split-tensile strength.

**Figure 7 materials-14-02156-f007:**
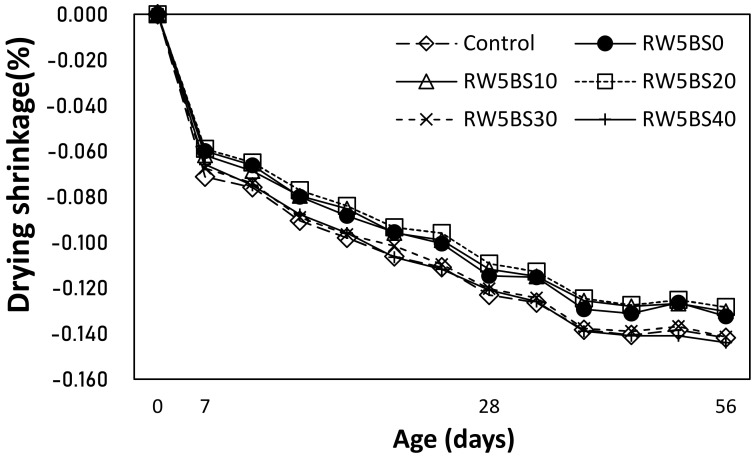
Drying shrinkage.

**Figure 8 materials-14-02156-f008:**
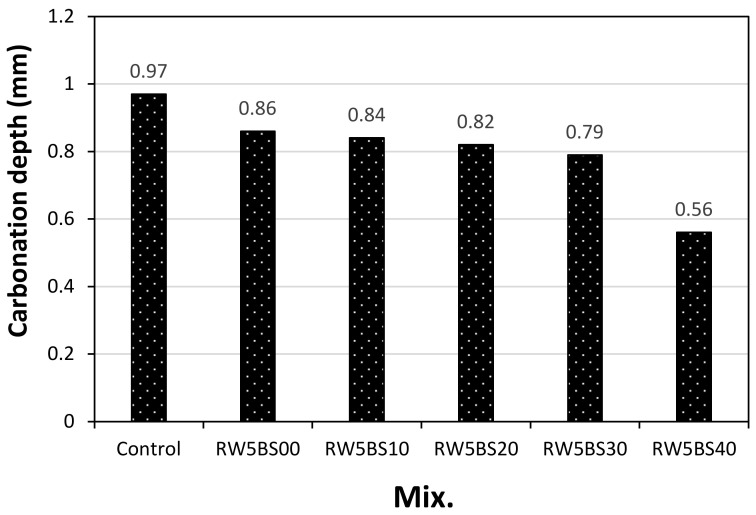
Accelerated carbonation depth.

**Figure 9 materials-14-02156-f009:**
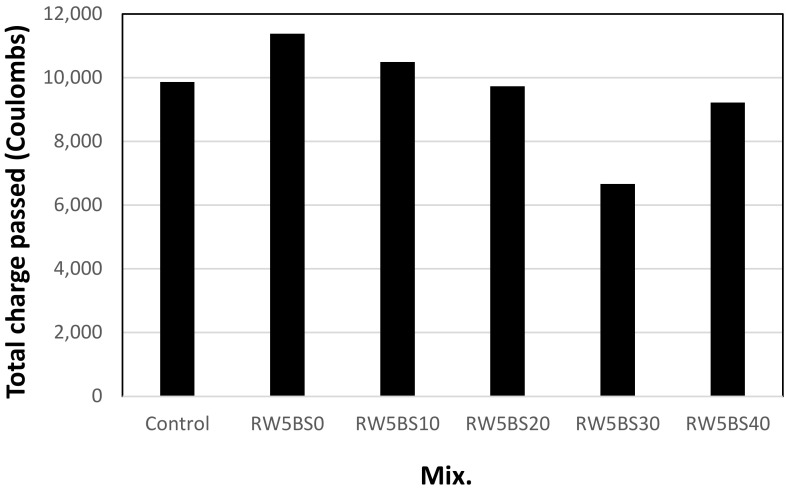
Chloride ion penetrability.

**Table 1 materials-14-02156-t001:** Sludge composition.

Mix	W/C (%)	Water (g)	Cement (g)	Sand Fines (g)
Sludge	50	200	400	100

**Table 2 materials-14-02156-t002:** Chemical composition of cement.

Type	SiO_2_	Al_2_O_3_	Fe_2_O_3_	CaO	MgO	K_2_O
OPC	17.43	6.50	3.57	64.40	2.55	1.17

**Table 3 materials-14-02156-t003:** Physical properties of fine aggregates.

Type	Fineness Modulus (FM)	Density	Water Absorption Ratio (%)
Natural fine aggregate (NFA)	2.45	2.60	1.0
Blast furnace slag fine aggregate (BSFA)	2.30	2.80	2.1

**Table 4 materials-14-02156-t004:** Mix proportion.

	BSFA (%)	Sludge Content (%)	W/C (%)	Water (kg/m^3^)	Cement (kg/m^3^)	NFA (kg/m^3^)	BSFA (kg/m^3^)
Control	0	0	50 170 340			739	0
RW5BS0	0	5				739	0
RW5BS10	10	5				665	80
RW5BS20	20	5				591	160
RW5BS30	30	5				518	240
RW5BS40	40	5				444	320

## Data Availability

Not applicable.

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
