# Peer review of "Strength and Durability Characteristics of Cement Composites with Recycled Water and Blast Furnace Slag Aggregate"

_materials, 2021, doi:10.3390/ma14092156_

Round 1

Reviewer 1 Report

The article is well prepared and addressing current topic of HPC concrete with reused water (RW) and blast furnace slag (BSFA) replacement instead of fine aggregate. The experiment is well designed with two control mixtures: a) OPC only; b) OPC and RW without the BSFA replacement. The articles draw important finding related to RW application in combination with BSFA with respect to strength, mortar flow and durability represented by carbonation test and chloride ion penetration resistance. The worse performance of the RW & BSFA mixtures at 7 days is expect able in my opinion. Since the process of hydration is extended with BSFA. Therefore, the BSFA mixtures over perform control mixture at 28 days as well as 56 days. It is worth mentioning that the RW related research brings novelty to the submitted manuscript. Questions and comments: 1) The mix proportions shall be clarified with respect to admixtures. Comments if there were the admixtures used (superplasticizers, high water range reducers etc.) or not is expected near the Table 3. 2) The typical number of samples is three per one test. T Comments about the number of samples is expected near the Table 3. 3) Are the: a) effect of the overheating during the ASTM C1202 called as the joule effect (see e.g. [1]), b) effect of chemical composition on the RW (alkalinity) and BSFA on the passed charge of ASTM C1202 (see e.g. [2]), considered? Note in the article related to those questions is expected at the page 7. References: [1] Ghosh, P, Hammond, A, and Tikalsky, P. J. “Prediction of Equivalent Steady State Chloride Diffusion Coefficients”, ACI Materials Journal, V. 108, No. 1, January – Feb, 2011, p. 88-94. [2] Pinhai Gao, Jiangxiong Wei, Tongsheng Zhang, Jie Hu, Qijun Yu "Modification of chloride diffusion coefficient of concrete based on the electrical conductivity of pore solution", Construction and Building Materials, V. 145, 2017, pp. 361-366, https://doi.org/10.1016/j.conbuildmat.2017.03.220. Formal: 4) There is redundant text on the page 2. I believe that is a from the template. Deleting of the text is required. The file with the comments is attached.

Author Response

Thank you for your comments to the manuscript.

Authors have revised the manuscript as below in accordance with reviewer’s comments.

We would like to resubmit this paper, and we hope you will consider this paper as suitable for publication in your journal.

We are looking forward to your reply.

Thank you in advance.

Reviewer 2 Report

The manuscript attempts to address the significance of using recycled wash water and blast furnace slag aggregates to minimize waste and address the ever-increasing resources scarcity in construction. At the same time, it examines the additional benefits of using BSFA along with recycled water. There is sufficient work put together but described poorly and shallowly. Besides, it is an informative article. My comments are listed as follows;

  1. The introduction should cover more literature in the area
  2. Line #66-81, needs to be deleted
  3. Line #83, under 2.1. Materials, it is better to have the PSD of both natural and BSFAs. The properties and chemical composition of the wash water need to be described
  4. Line #157, how does the alkalinity of recycled water relate to the reactivity of BSFA? You may also site any relevant literature
  5. Under 3.2 Compressive strength, results are not supported with enough reasoning.
  6. Under 3.3. Split-tensile strength, how does the increase in BSFA increases the tensile strength of the mortar? Can you put a little effort into the explanation? Think of correlating to shape, hydration, w/b ratio, particle packing, and amount of fines in the RW or other factors.
  7. Line #212, you have mentioned “…simultaneous use of BSFA and RW in cement mortar is expected to increase the chloride ion penetration resistance of the sample.” Have you tried to find a reason behind or any literature related to this?
  8. Generally, your arguments need to be supported with a reliable reason behind them. Try to correlate your findings with the existing literature. As an author, you need to explain why the findings are interesting and how they affect the understanding of the topic.

Author Response

(The authors gave the same response as above.)

Reviewer 3 Report

In this study, blast furnace slag fine aggregate (BSFA) and recycled water (RW) were used to make cement mortar. Many performance tests were conducted. In general, this is a well-organized paper and can be of general interest to the readers of Materials. The conclusions are well supported by the experimental results. However, there are some problems the authors should look into.

  1. The literature review is not sufficient.  The use of waste materials in cement and concrete industry is a big topic. The authors should give a bigger picture of this paper before focusing on slag and waste water. The recently published papers about waste tire and waste glass may help. ("Possibility of using waste tire rubber and fly ash with Portland cement as construction materials. Waste management29(5), pp.1541-1546."; "Cementless controlled low-strength material (CLSM) based on waste glass powder and hydrated lime: Synthesis, characterization and thermodynamic simulation. Construction and Building Materials275, p.122157." etc.)
  2. What is the limitation to the current understanding of the problem? What is the innovation of this study?
  3. In the last paragraph of Introduction, the authors should describe the objective of this research and the major experimental methods.
  4. What is the loading rate of strength test?
  5. The authors should discuss the results based on pozzolanic reaction between slag and cement.
  6. More in-depth discussions should be made to explain the possible underlying mechanism of this study. 

Author Response

(The authors gave the same response as above.)

Round 2

Reviewer 2 Report

The work is worth publishing but It lacks details in reasoning for the experimental results. It seems you prefer to leave it open and do more research.   

Reviewer 3 Report

This paper has been revised based on the comments.